# Plasma-Initiated Grafting of Bioactive Peptide onto Nano-CuO/Tencel Membrane

**DOI:** 10.3390/polym14214497

**Published:** 2022-10-24

**Authors:** Tzer-Liang Hu, Guan-Yu Chen, Shih-Chen Shi, Jason Hsiao Chun Yang

**Affiliations:** 1Division of General Neurosurgery, Jen-Ai Hospital, Taichung 41265, Taiwan; 2Department of Fiber and Composite Materials, Feng Chia University, Taichung 40724, Taiwan; 3Department of Mechanical Engineering, National Cheng Kung University, Tainan 70101, Taiwan

**Keywords:** Tencel, copper oxide, bioactive peptide, antimicrobial, atmospheric pressure plasma processing

## Abstract

A bioactive peptide has been successfully grafted onto nano-CuO impregnated Tencel membranes by a simple and rapid method involving a series of textile processes, and an atmospheric argon plasma treatment that requires no additional solvent or emulsifier. Surface morphology shows an apparent change from smooth, slightly roughened, and stripped with increasing plasma treatment time. The FT-IR characteristic peaks confirm the presence of the CuO nanoparticle and peptide on the extremely hydrophilic Tencel membranes that exhibit a zero-degree contact angle. Prepared nano-CuO/Tencel membranes with 90 s plasma treatment time exhibit excellent antimicrobial activity against *E. coli* and *S. aureus*, and promote fibroblast cell viability with the assistance of a grafted bioactive peptide layer on the membrane surface.

## 1. Introduction

As we enter yet another era of global conflicts, wounds are an unfortunate and an inevitable consequence from the ruthless battlefield. In such an event, immediate medical attention becomes absolutely necessary to ease inextricable physiological pain and emotional pain [1]. The high morbidity and mortality related to combat wounds are mainly due to uncontrolled hemorrhage and bacterial infections. Hemostatic and antibacterial characteristics, therefore, become a fundamental prerequisite for wound dressing materials [2,3].

From outdated medical cotton gauze to highly sophisticated moisture-balanced dressings, various types of dressings, including hydrogels [4,5], tissue adhesives [6,7,8], foams [9,10], alginates [11,12], and silicone-based materials [13,14], have been investigated to determine the optimum conditions in the wound healing processes [15,16]. Certain characteristics are needed for improved healing processes, which includes moisture-control, air permeability, exudate removal, antimicrobial, mechanical stability/elasticity, and biocompatibility/biodegradability [16]. In this study, Tencel, also known as lyocell, is used. It is known as an environment-friendly regenerated cellulose fiber used in textile research to enhance comfortable and breathable sensations [17,18]. It is also commonly adopted by the textile industry as a preferred material for fabricating garments such as sportswear, underwear, outerwear [18], and IoT-enabled smart clothing [19,20]. Over the past decade, extensive studies on moisture-wicking [21], softness [22], and enhanced mechanical behaviors [23] in Tencel have also opened up potential applications in wound dressings [24,25]. Tencel can not only fulfill most of the characteristics mentioned in the prior description, it is also cost effective, which makes production possible.

Tencel, however, has a drawback in its antimicrobial characteristics, which is usually expected in cellulose-based materials. In order to address this issue, researchers with pharmaceutical knowledge would likely adopt the idea of using antibiotics, such as ciprofloxacin [26], gentamicin [27], and mupirocin [28], as an antimicrobial additive. The incorporation of antibiotics can effectively deliver high concentrations of microbe-specific substances to local infectious sites, as suggested by Ramasubbu et al. [29]. However, there are cases where high amounts of antibiotics can cause long-lasting neurological damages [30]. Material scientists, on the other hand, are keener toward the use of different sorts of nanoparticles, such as silver [31,32,33,34], gold [35,36,37], metal oxides [38,39,40], and other metal-based compounds [41,42], to prevent bacterial infections and to overcome antimicrobial resistance (AMR) caused by the overuse and misuse of antibiotics.

To further reinforce the biofunction of Tencel, a bioactive agent has been implemented, as inspired by many studies on the strategic design of recent wound dressings [43]. It is evident that bioactive agents can induce or stimulate re-epithelialization on wound sites [44,45]. Although theoretical mechanisms are yet to be proven, many explanations have shown a certain depth of correlation with cell regulation on migration [46] and proliferation [47]. Other functions, such as anti-inflammatory from plant-based substances, have also been reported to be a potent tactic on epithelialization [48,49,50,51,52]. Researchers also attempted to alleviate the inflammatory response and support the healing process with different types of vitamins [53,54,55,56].

An ultimate challenge in the process of adding antimicrobial or bioactive agents would be the presence of more solvents or emulsifiers that could jeopardize the construction of suitable biocompatible scaffolds, and sacrifice the intended functionalities in various fabrication techniques. Eulálio et al. showed that the use of different organic acids has great impact on the physicochemical properties of chitosan [57]. Electrospun polymer-titanium dioxide nanocomposites, prepared by Ghosal et al., revealed the critical point for dispersion of nanoparticles, polymer spinnability, and the suitable application in biomedicine [58]. In addition, Fahimirad et al. demonstrated the use ethanol, methanol, and acetic acid for preparing multi-component, multi-layered structure membranes with antibacterial, antioxidant activity, and cytotoxicity effects on human epidermal cells [59].

Herein, the fabrication, characterization, and biological properties of peptide-coated nano-CuO/Tencel membranes for potential application in treating wounds are reported. The effects of atmospheric argon plasma treatment time on bioactive peptide coating on Tencel fibers are also evaluated. The prepared samples were characterized by scanning electron microscopy (SEM), Fourier-transform infrared (FT-IR) spectroscopy, contact angle measurement, and a variety of in vitro cell assays and antimicrobial assessments against skin-infection-related bacteria: Gram-negative *E. coli* and Gram-positive *S. aureus*.

## 2. Materials and Methods

### 2.1. Materials

Tencel^®^ fibers (purchased from Asiatic Fiber Corporation, Taipei, Taiwan) and highly concentrated 800-ppm proprietary bioactive peptide (acquired from Taiwan Goodwill Murray Peptide Technologies, Inc., Taipei, Taiwan) were used as received without further purification. Copper oxide nanoparticles were prepared by a green and rapid synthesis method, described in our previous work, such that zero hazardous substances were used or generated from the synthesis process [60].

### 2.2. Membranes Fabrication

A schematic illustration of the peptide coated nano-CuO/Tencel membrane fabrication is shown in Figure 1. Tencel fibers underwent a series of fiber processing from blending, carding, and needle-punching to produce a nonwoven Tencel sheet (basis weight: 100 g/m^2^) [61]. The nonwoven sheet was then immersed in deionized water with 4 wt.% copper oxide nanoparticle suspension, for optimal particle adhesion and an antimicrobial effect, onto Tencel fiber surface, followed by drying at room temperature for 4 h. A modified atmospheric pressure plasma jet system with a voltage of 45 kV, argon flow rate of 5 SLM, and a peptide injection flow rate of 0.05 L/min was used to coat bioactive peptide onto the nano-CuO/Tencel surface, where the tip of the plasma jet was placed 2 cm above the Tencel sheet. The peptide treatment time was set at 0, 30, 60, 90, and 120 s. The samples were kept in a desiccator up to three days before performing characterizations.

### 2.3. Scanning Electron Microscopy (SEM)

The surface morphology of the fabricated nano-CuO/Tencel with bioactive peptide coating was characterized. The samples were sputter-coated with gold, and viewed under a Hitachi (S-3400N, Tokyo, Japan) scanning electron microscope (SEM) at an accelerating voltage between 5 and 20 kV, and mounted with energy dispersive X-ray analysis (EDX) in the Precision Instruments Support Center, Feng Chia University. Micrographs were collected at magnification of 1k× and 5k× using secondary electrons and backscattered electrons.

### 2.4. Fourier-Transform Infrared Spectroscopy (FT-IR)

Transmission Fourier-transform infrared (FT-IR) spectroscopic measurements were performed on a Fourier-transform infrared spectrophotometer (Thermo Nicolet iS5 FTIR, Waltham, MA, USA) in National Cheng Kung University. All the FT-IR measurements were repeated three times for each sample, and are well reproducible.

### 2.5. Contact Angle Measurements

The contact angle was measured using a static method and direct measurement of the tangent angle, *θ*, at the three-phase contact point on a sessile drop profile, using high-resolution photographs of pure water drops and a graphic processing software supplied by the manufacturer of the contact angle meter (CAM-100, Creating Nano Technologies Inc., Tainan, Taiwan). When *θ* is higher than 90°, the surface is hydrophobic, whereas it is hydrophilic when the angle is less than 90°.

### 2.6. Cytotoxicity Evaluation

L929 standard fibroblasts (ATCC cell line) were cultured in Dulbecco’s modified Eagle’s medium (DMEM) and supplemented with 10% fetal bovine serum (FBS) and 1% penicillin-streptomycin. Cells were maintained at 37 °C in a humidified incubator with 5% CO_2_ for 24 h, until about 80% confluence was obtained. Cell counts were standardized.

The prepared membranes (n = 3) were placed into 24-well tissue culture polystyrene plates without further treatment. After UV irradiation sterilization, cells were seeded at a density of 1 × 10^5^ cells/cm^2^ in each well plate. Cellular viability was assessed using a 3-(4,5-dimethylthiazol-2-yl)-2,5-diphenyltetrazolium bromide (MTT) assay taken after 24 h. The culture medium was discarded, followed by washing with PBS twice and incubation with MTT solution at 37 °C for 4 h. The formazan crystals were dissolved in dimethyl sulfoxide (DMSO), and optical activity was measured at 570 nm with an ELISA reader.

### 2.7. Antimicrobial Assay

The inhibitory effects of copper-tailored membranes on the bacterial growth were estimated by means of turbidity measurements. The reference bacterial broth (3 × 10^8^ bacteria/mL) was used as a standard sample. Selected membranes (1 cm × 1 cm) were cultured in *E. coli* (six-fold serial dilution) and *S. aureus* (four-fold serial dilution) bacterial broth at room temperature for 24 h. The optical density (OD) of all solutions was measured with an UV/VIS spectrophotometer (JASCO V-550) at 620 nm.

## 3. Results and Discussion

### 3.1. Surface Morphology of Nano-CuO/Tencel Membranes

The surface morphology of the prepared Tencel fibers were studied by SEM. This technique enabled us to examine the effect of plasma treatment on a single fiber in the fabrication sequence. Plasma treatment resulted in extensive changes of the Tencel surface morphology. Figure 2 shows SEM images of pristine Tencel, CuO-coated Tencel, and peptide-included CuO/Tencel fibers with different plasma treatment times. Pristine Tencel fibers appear to have a smooth fiber surface with an average diameter around 18 μm (Figure 2a). As shown in Figure 2b–g it is revealed that fiber diameters ranging from 15–22 μm are observed in Tencel fibers impregnated with CuO particulate, whereas no significant change in fiber diameter is observed for peptide-coated fibers (Figure 2c–g). Apparent CuO nanocrystals can be seen in Figure 2b (indicated with an arrow), with the inset showing elemental values in percentage. No noticeable change is observed when increasing the argon plasma treatment time from 0 to 60 s (Figure 2c–e); however, a slightly roughened surface morphology becomes visible with 90 s exposure time (Figure 2f). When plasma exposure time is further extended to 120 s, severe stripping on the fiber surface is detected in Figure 2g, where evidence of partial deterioration is clearly seen in the close-up view (Figure 2h). The observed surface changes and stripped fibers are the consequence of sample ablation during plasma treatment [62,63,64]. The fibers create a greater surface area in comparison to the smooth surface of pristine Tencel fibers. As reported by Lewis et al., larger surface area is responsible for superior hemostatic properties [65].

### 3.2. FT-IR Analysis of Nano-CuO/Tencel Membranes

The FT-IR spectra for nano-CuO/Tencel membranes with different plasma exposure times were recorded in the range of 4000–400 cm^−1^. As shown in Figure 3, the FT-IR spectra of Tencel fabric show distinctive peaks, including O-H stretching at 3399 cm^−1^, C-H stretching at 2985 cm^−1^, O-H bending at 1633 cm^−1^, CH_2_ bending at 1427 cm^−1^, and the C-H bending at 1387 cm^−1^ [66]. These peaks are the characteristic peaks of cellulose. The observed peaks at 453, 494, and 609 cm^−1^ (in dashed circle) correspond to the characteristic stretching vibrations in CuO [67]. It is observed that such IR feature is not particularly distinctive when peptide is added. Generally speaking, the absorbance in the amide region has a strong correlation to the amino side chain in collagen or peptide [68,69]. It is assumed that the absorbance of 1635 cm^−1^ mainly arises from amide contribution [70]. This spectral feature is particularly obvious in the Tencel+CuO+Peptide_30s-90s samples. In samples treated with plasma, the band at 1749 cm^−1^ corresponds to -C=O in -COOH. The presence of carboxyl groups in these spectra is the result of plasma treatment [71]. According to the results, the amino groups of the peptide forms the Schiff-base structure onto the Tencel fiber surface, which aligns with the IR absorbance at 1635 cm^−1^ (Figure 4) [72]. As the fibers are progressively etched (Tencel+CuO+Peptide_120s), the amide I and -C=O bands become reduced or diminished. In addition, the strong absorption bands between 3500–4000 cm^−1^ and 1624 cm^−1^ (overlapped with O-H bending) are allocated to the presence of water molecules that are absorbed on the high surface-to-volume ratio nanostructures [73,74].

### 3.3. Cytotoxicity and Contact Angle Measurements of Nano-CuO/Tencel Membranes

An effective wound dressing material must be non-cytotoxic to the relevant cell to maintain viability at the wound site. To reveal the cyto-compatibility of the tested membranes, the cell viability of the L929 fibroblastic cells was investigated. The L929 cell line is one of the most frequently used lines in material/cell interaction research, and has been previously used for cytotoxicity testing for many polymeric scaffolds [75,76,77]. The cytotoxicity and wettability of the prepared membranes are summarized in Figure 5. Tencel is extremely hydrophilic, and the high wetting behavior is primarily attributed to the -OH group from the cellulose-based structure (blue circles in Figure 5) [73]. Moreover, Tencel fibers pose no adverse effect on fibroblastic cells, such that the cell viability is very similar to that of the control sample. In Tencel+CuO, CuO nanoparticles appear to have a high cytotoxicity, whereas peptide-added samples appear to have little or no cytotoxic effect. Others also demonstrated a similar outcome that copper(II) complex or copper oxide nanoparticles have toxicity against L929 mouse fibroblast, A549 human lung cancer cells, or to DNA from reactive oxygen species (ROS), which enables us to comprehend apoptosis induction or an anti-proliferation effect of CuO in Tencel membranes [78,79,80]. Fibroblasts (NIH/3T3) induced with bioactive peptide show a faster gap closure rate (Figure A1A), whereas the quantitative result reveals a slower cell migration without peptide treatment (square curve) compared to peptide-treated cells (circle curve). A significant split in wound area difference is seen at 6 h, and reaches its maximum at around 13 h where the wound area is almost closed (Figure A1B). The assessment, however, was not performed on the peptide-CuO/Tencel samples because of technical limitations. The effects of the peptide-grafted Tencel samples on cell viability can be associated and explained by the degree of peptide adhesion from plasma treatment according to the IR results from Tencel+CuO+Peptide_30s-90s samples. 

### 3.4. Antimicrobial Activity

One of the pivotal factors to accelerate wound healing is the ability to impede microbial infections [81,82]. Therefore, copper oxide nanoparticles were introduced as an inorganic antimicrobial agent. The antimicrobial behaviors of the devised nano-CuO/Tencel membranes against *E. coli* and *S. aureus* were evaluated as optical density (OD) measurements, as presented in Figure 6. The antimicrobial nature of copper oxides is also clearly observed against both Gram-positive and -negative bacterial strains [83]. For both bacterial solutions exposed to nano-CuO-coated Tencel surfaces, the absorbance readings are reduced, indicating that these surfaces can inhibit bacteria growth in a liquid medium. The inhibitory action is more obvious for Gram-negative bacteria, as the overall absorbance in Gram-positive bacteria is much higher. The results suggest an electrostatic interaction between positively charged copper ions that disrupts the membrane integrity of the Gram-negative bacteria [84]. As expected from the cytotoxicity test, the presence of peptide also shows similar patterns for plasma-treated samples in the antimicrobial results. As peptide is being applied onto the nano-CuO/Tencel fibers, argon ions are initiating reactive species for reaction and etching newly established bonds simultaneously [85]. Thus, the absorbance readings for both bacteria begin to decline at 90 s plasma treatment time, mainly because of the competing result of grafting and removing of peptide that leads to the protrusion of copper oxide for antimicrobial activity. The stripped fibers at 120 s of plasma treatment time ultimately deteriorated, and demonstrated unstable and inconsistent antimicrobial capability.

## 4. Conclusions

An innovative method to graft bioactive peptide onto nano-CuO/Tencel membranes has been proposed. The results have revealed that membranes remain high fiber surface integrity between 30 and 60 s of plasma treatment time. Though slight roughness is observed at 90 s that etches off peptide partially to promote antimicrobial activity against *E. coli* and *S. aureus*, cell viability with fibroblastic cells is still maintained. Ongoing experiments, including peptide adhesion on the fiber surface, and its cellular response mechanism, shall be carefully investigated for future publication.

## Figures and Tables

**Figure 1 polymers-14-04497-f001:**
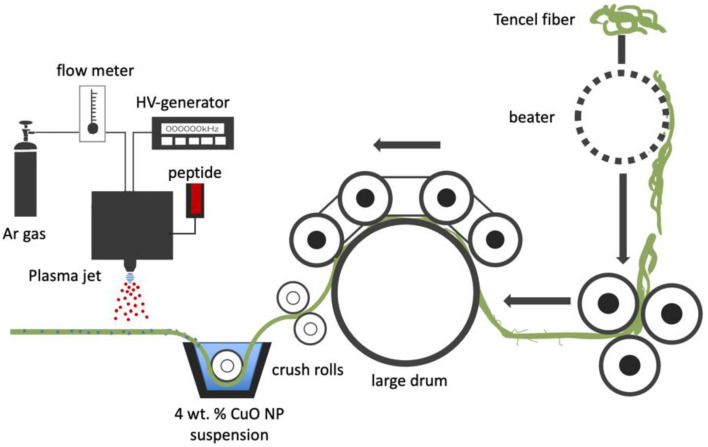
Schematic of the fabrication sequence of peptide-coated nano-CuO/Tencel.

**Figure 2 polymers-14-04497-f002:**
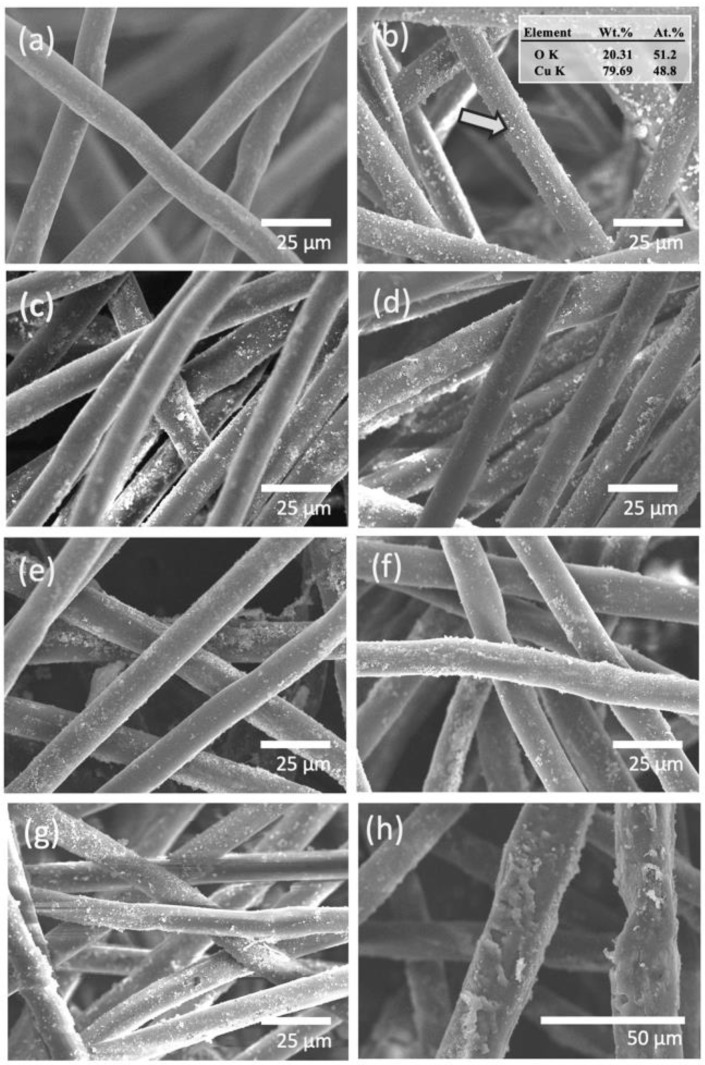
Scanning electron microscopy images of (**a**) Tencel; (**b**) Tecenl+CuO; and Tecenl+CuO+peptide (**c**) 0, (**d**) 30, (**e**) 60, (**f**) 90, and (**g**,**h**) 120 s of argon plasma treatment time at low and high magnification.

**Figure 3 polymers-14-04497-f003:**
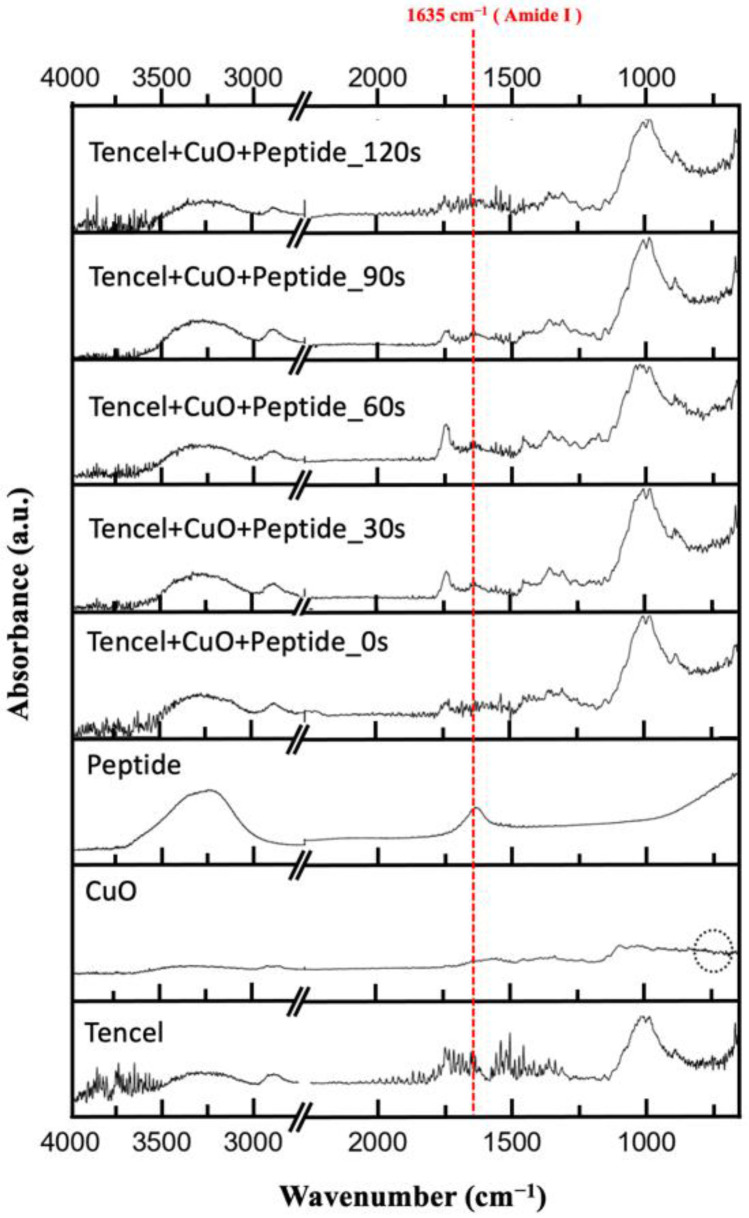
FT-IR spectra of nano-CuO/Tencel membranes with different plasma treatment times.

**Figure 4 polymers-14-04497-f004:**
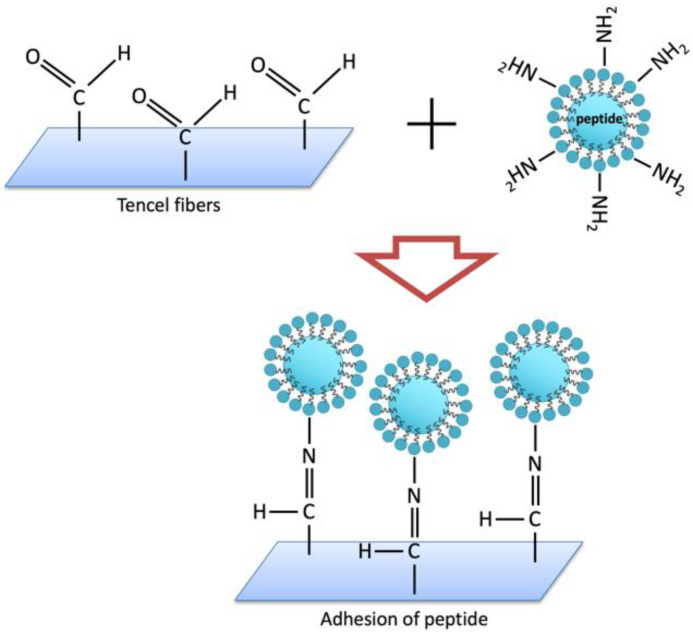
Schematic illustration of formation mechanism of peptide onto Tencel membranes.

**Figure 5 polymers-14-04497-f005:**
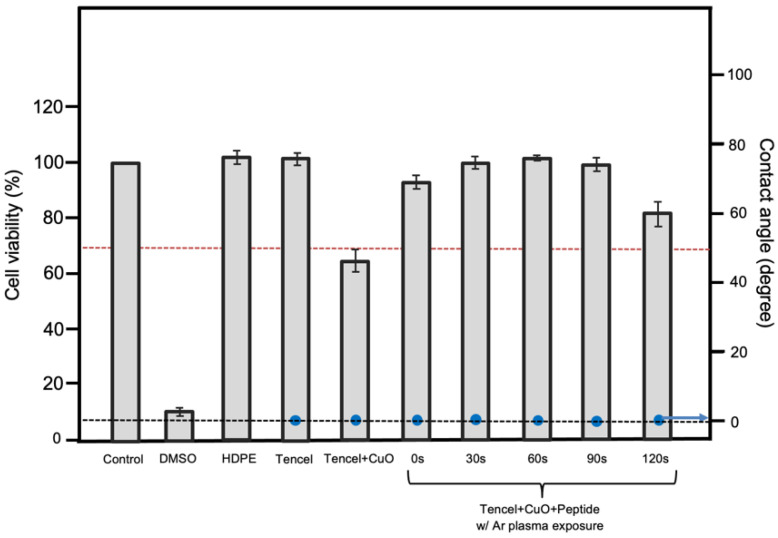
Cytotoxicity (red dashed line indicates 70% viability) and contact angle measurement (black dashed line indicates zero degree) of the Tencel membranes.

**Figure 6 polymers-14-04497-f006:**
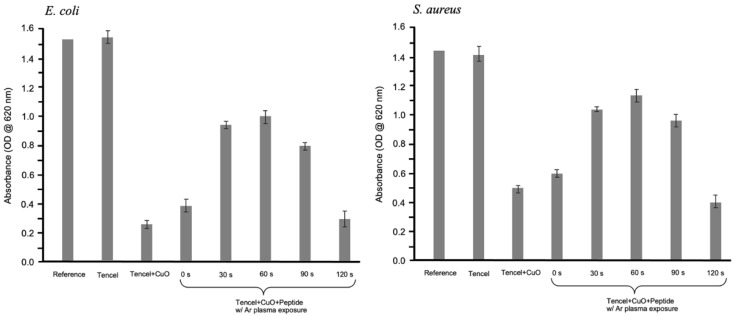
Absorbance, as measured at 620 nm, of *E. coli* (**left**) and *S. aureus* (**right**) bacterial broth exposed to various membranes.

## Data Availability

Not applicable.

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
