# Peer review of "Plasma-Initiated Grafting of Bioactive Peptide onto Nano-CuO/Tencel Membrane"

_polymers, 2022, doi:10.3390/polym14214497_

Round 1
Reviewer 1 Report
The manuscript polymers-1792714 entitled; “Plasma-Initiated Grafting of Bioactive Peptide onto Nano-CuO/Tencel Membrane”. The authors assess an innovative method to graft bioactive peptides onto nano-CuO/Tencel membranes has been proposed. Results have revealed that membranes remain high fiber surface integrity between 30 and 60 sec of plasma treatment time. The main findings of the available data were there was slight roughness observed at 90 sec that etches off peptide partially to promote antimicrobial activity against E. coli and S. aureus while maintaining acceptable cell viability with fibroblastic cells. I have no comments regarding the protocol itself, or the organization of the manuscript. In my opinion, the article in the presented version is suitable for publication after minor revisions.
- Please check all abbreviations throughout the manuscript.
- In the introduction section was too long, please use more recent references related to your work.
Author Response
Reviewer: 1
The manuscript polymers-1792714 entitled; “Plasma-Initiated Grafting of Bioactive Peptide onto Nano-CuO/Tencel Membrane”. The authors assess an innovative method to graft bioactive peptides onto nano-CuO/Tencel membranes has been proposed. Results have revealed that membranes remain high fiber surface integrity between 30 and 60 sec of plasma treatment time. The main findings of the available data were there was slight roughness observed at 90 sec that etches off peptide partially to promote antimicrobial activity against E. coli and S. aureus while maintaining acceptable cell viability with fibroblastic cells. I have no comments regarding the protocol itself, or the organization of the manuscript. In my opinion, the article in the presented version is suitable for publication after minor revisions.
- Please check all abbreviations throughout the manuscript.
Answer: Thank you for your comments. Abbreviations in the manuscript have been reviewed.
- In the introduction section was too long, please use more recent references related to your work.
Answer: Thank you for your comments. In our initial submission, an editorial member suggested us to provide adequate background on this particular study, and thus we included in-depth background from different aspects. However, we do agree with you that the introduction section is lengthy.
Reviewer 2 Report
The authors have coated Tencel Fibers with CuO nanoparticles and then grafted bioactive peptides onto the fibers. Thanks to the authors for their hard work and dedication. I have some queries and suggestions which are mentioned below:
1. This article reported grafting of bioactive peptide but there is no evidance of peptide bond present in FT-IR data then how authors can claim that peptide has been grafted?
2. There is no description how CuO nanoparticles were attached with tencel fibers.
3. If peptide is grafted on CuO/Tencel fibers then which one is mainly responsible for antibacterial performance? Peptide or CuO?
4. Section 2.2: Authors are requested to describe membrane fabrication process step by step. The schematic diagram in figure 1 is confusing. For example, “The nonwoven sheet was then immersed in a 4 wt.% copper oxide nanoparticle suspension in deionized water, ….."
well! Then, how these sheets moved forward under the plasma jet? How Tencel Fibers were moved from large drum to CuO suspension?
6. It is better to mention the reason behind taking 4 wt.% CuO.
7. Does l/min (in line 99) indicate Litter/minute? If so then please use capital L to indicate litter.
Author Response
The authors have coated Tencel Fibers with CuO nanoparticles and then grafted bioactive peptides onto the fibers. Thanks to the authors for their hard work and dedication. I have some queries and suggestions which are mentioned below:
- This article reported grafting of bioactive peptide but there is no evidance of peptide bond present in FT-IR data then how authors can claim that peptide has been grafted?
Answer: Thank you for your comments. We have performed another FTIR analysis on the samples (pristine, CuO, peptide, and fabricated membranes plasma treated at 30, 60, 90, and 120 s) with improved technique to assess the presence of peptide. The updated results show an apparent absorbance of amide I band from peptide on plasma treated membranes.
- There is no description how CuO nanoparticles were attached with tencel fibers.
Answer: Thank you for your comments. The description has been revised to clarify the coating process of CuO onto Tencel membranes. The coating of CuO was carried out by immersing the Tencel membranes in deionized water with CuO nanoparticle suspension.
- If peptide is grafted on CuO/Tencel fibers then which one is mainly responsible for antibacterial performance? Peptide or CuO?
Answer: Thank you for your comments. As shown in the revised Fig. 5, the presence of calcium oxide contributes to the antibacterial effect on the CuO/Tencel membranes.
- Section 2.2: Authors are requested to describe membrane fabrication process step by step. The schematic diagram in figure 1 is confusing. For example, “The nonwoven sheet was then immersed in a 4 wt.% copper oxide nanoparticle suspension in deionized water, ….."
Answer: Thank you for your comments. The description has been revised to clarify the coating process of CuO onto Tencel membranes. The coating of CuO was carried out by immersing the Tencel membranes in deionized water with CuO nanoparticle suspension.
- well! Then, how these sheets moved forward under the plasma jet? How Tencel Fibers were moved from large drum to CuO suspension?
Answer: Thank you for your comments. The illustration is a representative figure of the processing sequence. The pristine Tencel fibers were immersed in CuO suspension, dried in room temperature, and exposed in atmospheric plasma for 30, 60, 90, and 120 seconds along with peptide injection.
- It is better to mention the reason behind taking 4 wt.% CuO.
Answer: Thank you for your comments. Experimentally, 4 wt.% CuO suspension in deionized water was preferred for improved particle attachment on the fiber surface.
- Does l/min (in line 99) indicate Litter/minute? If so then please use capital L to indicate litter.
Answer: Thank you for your comments. The flow rate has been changed to capital letter as suggested.
Reviewer 3 Report
Journal Title: Polymers
Manuscript Title: Plasma-Initiated Grafting of Bioactive Peptide onto Nano-CuO/Tencel Membrane
Manuscript ID: polymers-1792714
Authors: Tzer-Liang Hu, Guan-Yu Chen, Shih-Chen Shi, Jason Hsiao Chun Yang
The manuscript entitled “Plasma-Initiated Grafting of Bioactive Peptide onto Nano-CuO/Tencel Membrane” treats the subject of a cellulosic type membrane production being activated whit CuO NPs and biofunctionalized whit a peptide on the surface. The authors have performed structural and morphological characterization. The fabricated membranes have been tested regarding their cytotoxicity, wound-healing capacity and antimicrobial ability.
The overall scientific contribution of the paper in the field is of relatively good, the content can be interesting for the reader, the manuscript must yet be improved.
Major revision is required according to the following questions & recommendations:
-line 74, check “nano-Cu/Tencel membranes”, there is an O missing
-it is not clear how many samples were produced, is there a sample without CuO?
-is there a difference between the sample whit 0 sec plasma exposure and the sample containing CuO activated Tencel? Please revise, please refer to exact number of samples, give code if needed
-line 181, it is recommended to provide images at same magnification, the authors should choose between 25 or 50 μm; also, keep the dimension bar on all the images related to all the samples; again, pristine fibers do contain CuO or not? The presented text and figure 2 legend is ambiguous.
-line 201, sample having a plasma treatment time of 0 sec contain peptide or not? Please explain, please revise, is definitely unclear information; is there a ftir spectrum for the cellulose membrane without CuO, please provide it too
-line 192, “of Cu-O bond in CuO” should be revised…
-line 198, the sentence “No detectable peak was observed for the bioactive peptide due to relatively small amount presented in all the samples.” It is not satisfactory… at least the FTIR spectrum of the pure peptide should be provided. Comments about the functional groups of the peptide should be given. In this context, the paragraph from 98 to 101 lines becomes doubtful… is there a coating of peptide or not?… the author must quantify the used substances, please give more data… how much is “small amount”?
- authors have the ability to perform XPS? It will be enlightening
-more studies on the mechanism of interaction between the Ar plasma and peptide and the CuO activated and/or pristine cellulose must be provided in the revised manuscript
-no information about the purification protocol before cytotoxic measurements were provided, please fix this; same for the antimicrobial assay
-line 213, “The introduction of bioactive peptide, however, reverses the cytotoxic effect” The introduction of bioactive peptide was not observed by FTIR “due to relatively small amount presented in all the samples.” How now the cytotoxic effect of CuO NPs can be reversed? Please, explain
-Figure 5A, it is not clear which sample is the one treated with peptide? Is the sample of 0 sec exposure, is 30, 60, 90, 120?
-Figure 6 vs. figure 4: the sample CuO/Tencel proved to be cytotoxic. Why is this sample missing in the antimicrobial measurement? If CuO was used to induce AM effect? The data must be revised
-Unfortunately, the experiments including peptide adhesion on fiber surface, and its cellular response mechanism shall be carefully investigated before the publication of the current work
-the references seem too many, the most representative shall be kept
Author Response
The manuscript entitled “Plasma-Initiated Grafting of Bioactive Peptide onto Nano-CuO/Tencel Membrane” treats the subject of a cellulosic type membrane production being activated whit CuO NPs and biofunctionalized whit a peptide on the surface. The authors have performed structural and morphological characterization. The fabricated membranes have been tested regarding their cytotoxicity, wound-healing capacity and antimicrobial ability.
The overall scientific contribution of the paper in the field is of relatively good, the content can be interesting for the reader, the manuscript must yet be improved.
Major revision is required according to the following questions & recommendations:
-line 74, check “nano-Cu/Tencel membranes”, there is an O missing
Answer: Thank you for your comments. The mistake has been corrected.
-it is not clear how many samples were produced, is there a sample without CuO?
Answer: Thank you for your comments. Tencel, Tencel+CuO, Tencel+CuO+peptide (0, 30, 60, 90, sec plasma treatment time) were prepared to demonstrate the sequence of fabrication. Tencel and Tencel+CuO, however, are used as references to show cytotoxic effect and antimicrobial activities.
-is there a difference between the sample whit 0 sec plasma exposure and the sample containing CuO activated Tencel? Please revise, please refer to exact number of samples, give code if needed
Answer: Thank you for your comments. The referred samples (Tencel+CuO and Tencel+CuO+peptide_0s) are not the same. Please see the text and figures for updated description.
-line 181, it is recommended to provide images at same magnification, the authors should choose between 25 or 50 μm; also, keep the dimension bar on all the images related to all the samples; again, pristine fibers do contain CuO or not? The presented text and figure 2 legend is ambiguous.
Answer: Thank you for your comments. Magnification of 1k were used to capture all the fiber conditions, while 5kX was used for enlarged view of Tencel+CuO+peptide_120s.
-line 201, sample having a plasma treatment time of 0 sec contain peptide or not? Please explain, please revise, is definitely unclear information; is there a ftir spectrum for the cellulose membrane without CuO, please provide it too
Answer: Thank you for your comments. As described in the updated text, peptide is included in all the plasma treated samples. Text and corresponding figures have been corrected with updated information.
-line 192, “of Cu-O bond in CuO” should be revised…
Answer: Thank you for your comments. The sentence has been revised.
-line 198, the sentence “No detectable peak was observed for the bioactive peptide due to relatively small amount presented in all the samples.” It is not satisfactory… at least the FTIR spectrum of the pure peptide should be provided. Comments about the functional groups of the peptide should be given. In this context, the paragraph from 98 to 101 lines becomes doubtful… is there a coating of peptide or not?… the author must quantify the used substances, please give more data… how much is “small amount”?
Answer: Thank you for your comments. We have performed another FTIR analysis on the samples (pristine, CuO, peptide, and fabricated membranes plasma treated at 0, 30, 60, 90, and 120 s) with improved technique to assess the presence of peptide. The updated results show an apparent absorbance of amide I band from peptide on plasma treated membranes.
- authors have the ability to perform XPS? It will be enlightening
Answer: Thank you for your comments. At the present stage, XPS analysis is not included. We do hope that we will be able to include such characterization in the future to address the mechanism of peptide grafting.
-more studies on the mechanism of interaction between the Ar plasma and peptide and the CuO activated and/or pristine cellulose must be provided in the revised manuscript
Answer: Thank you for your comments.
-no information about the purification protocol before cytotoxic measurements were provided, please fix this; same for the antimicrobial assay
Answer: Thank you for your comments. The necessary information regarding the cytotoxicity and antimicrobial tests are written.
-line 213, “The introduction of bioactive peptide, however, reverses the cytotoxic effect” The introduction of bioactive peptide was not observed by FTIR “due to relatively small amount presented in all the samples.” How now the cytotoxic effect of CuO NPs can be reversed? Please, explain
Answer: Thank you for your comments. We have re-examined the samples (pristine, CuO, peptide, and fabricated membranes plasma treated at 30, 60, 90, and 120 s) with improved technique to assess the presence of peptide. With the updated results, we can conclude the reduced cytotoxic effect from CuO nanoparticles is the result of peptide grafting.
-Figure 5A, it is not clear which sample is the one treated with peptide? Is the sample of 0 sec exposure, is 30, 60, 90, 120?
Answer: Thank you for your comments. The description has been revised to clarify samples with or without peptide.
-Figure 6 vs. figure 4: the sample CuO/Tencel proved to be cytotoxic. Why is this sample missing in the antimicrobial measurement? If CuO was used to induce AM effect? The data must be revised
Answer: Thank you for your comments. The data have been revised.
-Unfortunately, the experiments including peptide adhesion on fiber surface, and its cellular response mechanism shall be carefully investigated before the publication of the current work
Answer: Thank you for your comments. The present work focuses on the preparation of Tencel membranes with CuO and peptide addition, and further assesses the cellular response and antimicrobial activities on the fabricated membranes. The cellular response mechanism, however, shall be further addressed and investigated in our future work.
-the references seem too many, the most representative shall be kept
Answer: Thank you for your comments. In our initial submission, an editorial member suggested us to provide adequate background on this particular study, and thus we included as many references as possible to support the development of our. However, we do agree with you that the number of references is excessive.
Round 2
Reviewer 2 Report
Thank you for responding all issues.
Reviewer 3 Report
Journal Title: Polymers
Manuscript Title: Plasma-Initiated Grafting of Bioactive Peptide onto Nano-CuO/Tencel Membrane
Manuscript ID: polymers-1792714_R1
Authors: Tzer-Liang Hu, Guan-Yu Chen, Shih-Chen Shi, Jason Hsiao Chun Yang
The authors have reviewed properly the initial manuscript which actually became a nice work ready to be published in Polymers.
Still, some minor typo errors are present (which may be fixed when proofing):
-In Abstract, “The FT-IR characteristic peaks at confirm the presence of CuO nanoparticle and peptide on the extremely hydrophilic Tencel membranes that exhibit zero-degree contact angle.”
-line 27, there is a space in between “pain [1]. The high”
-In Figure 2, “g” letter might be changed from black to white as others, for consistency
-line 178, please verify if the term “fibric” is legit
-line 209, check the meaning of “the cell viability is very similar to of the control sample”
-line 219, there is a space in between “(Fig. A1(b)). Such”
